# Current practices in studies applying the target trial emulation framework: a protocol for a systematic review

Theophile Bigirumurame ![ORCID] , Shaun Kuan Wei Hiu ![ORCID] , M Dawn Teare ![ORCID] , James M S Wason ![ORCID] , Andrew Bryant ![ORCID] , Matthew Breckons ![ORCID]

## ABSTRACT

**Introduction** Observational studies represent an alternative to estimate real-world causal effects in the absence of available randomised controlled trials (RCTs). Target trial emulation is a framework for the application of RCT design principles to emulate a hypothetical open-label RCT (the hypothetical target trial) using existing observational data as the primary data source as opposed to the prospective recruitment and measurement of randomised units. The aim of this systematic review is to investigate the practices of studies applying the target trial emulation framework to evaluate the effectiveness of interventions.

**Methods and analysis** We will systematically search in Medline (via Ovid), Embase (via Ovid, entries from medRxiv are included), PsycINFO (via Ovid), SCOPUS, Web of Science, Cochrane Library, the ISRCTN registry and ClinicalTrials.gov for all study reports and protocols which used the trial emulation framework (without time restriction). We will extract information concerning study design, data source, analysis, results, interpretation and dissemination. Two reviewers will perform study selection, data extraction and quality assessment. Disagreements between reviewers will be resolved by a third reviewer. A narrative approach will be used to synthesise and report qualitative and quantitative data. Reporting of the review will be informed by Preferred Reporting Items for Systematic Review and Meta-Analysis guidance (PRISMA).

**Ethics and dissemination** Ethical approval is not required as it is a protocol for a systematic review. Findings will be disseminated through peer-reviewed publications and conference presentations.

## STRENGTHS AND LIMITATIONS OF THIS STUDY

⇒ This systematic review protocol details a comprehensive search strategy consisting of academic databases, Cochrane library and academic repositories and archives.
⇒ This systematic review protocol pre-specifies qualitative research methods to synthesise extracted text data.
⇒ This systematic review protocol follows the Preferred Reporting Items for Systematic Review and Meta-Analysis Protocols guidelines.
⇒ Risk of bias assessments will not be included in the planned systematic review.
⇒ Exclusion of non-English electronic databases may introduce language bias.

Population Health Sciences Institute, Newcastle University, Newcastle upon Tyne, UK

**Correspondence to**
Dr Theophile Bigirumurame;
theophile.bigirumurame@
newcastle.ac.uk

## INTRODUCTION

Observational studies represent an alternative to estimate real-world causal effects in the absence of available randomised controlled trials (RCTs). Undoubtedly, RCTs are positioned toward the top of the evidence hierarchy in the evaluation of the efficacy and safety of treatments. However, reasons to conduct observational studies instead of RCTs may include better external validity, feasibility, ethical considerations, reduced associated costs and greater time efficiency.[1 2] While the use of observational studies to attempt to estimate causal effects is not a new phenomenon—the practice of which dates back as early as Rubin's work[3] on counterfactual theory—the application of RCT design principles to overcome biases in and improve the quality of observational research has been a relatively new development.

Target trial emulation is a framework for the application of RCT design principles to emulate a hypothetical open-label RCT (the hypothetical target trial) using existing observational data as the primary data source as opposed to the prospective recruitment and measurement of randomised units.[4] Because treatment assignment is neither blind nor random, valid causal treatment effects are estimated if the identification principles of causal inference are satisfied.[4–8] These include the assumptions of exchangeability (participants in a study may be swapped between treatment and control groups without changing the average treatment effect), consistency (no multiple versions of the treatment options being investigated), positivity (every patient has a non-zero probability to receive every treatment option under investigation given their measured covariates) and non-interference (the treatment given

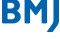

to one patient does not affect the outcome of another patient).[7 9–13] A table of acronyms, abbreviations and technical terms are defined in online supplemental appendix.

A study employing target trial emulation is characterised by an explicit description of the hypothetical target trial on several design components: the eligibility criteria, treatment strategy, assignment procedure, follow-up, primary outcome definition, the causal contrast of interest and a statistical analysis plan.[4] Additionally, an accompanying target trial emulation specification is described and makes explicit how the hypothetical target trial is being emulated along the aforementioned design elements. Stating both the hypothetical target trial and the target trial emulation specifications has important benefits. It assists investigators in avoiding biases and pitfalls that are common to the analysis of observational data[14 15] and permits the sceptical reader to judge if the design of the hypothetical target trial was sensible and if emulation was satisfactory; for example, the eligibility criteria may not be faithfully emulated owing to limitations with the observational data.

Target trial emulation studies can be credible sources of real-world evidence (RWE) to support regulatory decisions in the absence of RCTs, for instance, within the Food and Drug Administration's Real-World Evidence Program[16] and the European Medicines Agency's OPerational, TechnIcal, and MethodologicAL framework (OPTIMAL).[17] Furthermore, with the increasing availability of large-scale real-world data (RWD) especially those seen in research cohorts, registry, hospital and general practitioner databases, there is impetus for increased popularity of the application of target trial emulation.

To our knowledge, no study has reviewed the extant work applying the target trial emulation framework. We argue that a review of this nature is needed for several reasons that have implications for the continued growth of target trial emulation.

First, a systematic review (SR) that reports on the design, conduct and reporting of target trial emulation studies will help facilitate new research hoping to use the methodology. Investigators seeking to conduct their own target trial emulation study may draw guidance from what is currently published, and so an SR of the literature serves as a convenient repository of knowledge to new investigators who may be unfamiliar with terminology, choices of designs and analyses, and reporting practices. An additional benefit of the review to both new and experienced investigators is the increased awareness of available datasets that may be pertinent to their field of expertise.

Second, target trial emulation studies attempt to mimic RCTs and thus the Consolidated Standards of Reporting Trials (CONSORT) checklist[18] may be a valid guideline for reporting standards. However, given that target trial emulation studies are necessarily observational studies, the Strengthening the Reporting of Observational Studies in Epidemiology (STROBE) guidelines also apply.[19] Thus, it is essential to capture what (or whether) established reporting guideline(s) have been adopted within the existing trial emulation literature, so that future studies may follow suit. Furthermore, with the recent movement towards the language of the estimand framework in clinical trials,[20] it may be useful to capture whether the target trial emulation literature has also displayed such a shift. It may also be that existing checklists are not sufficient to capture all the nuances that exist in target trial emulation. For example, explicit statements on quality assessment such as how the data have been evaluated and judged to be fit-for-purpose for trial emulation are not expected by existing guidelines. Thus, any available reports on data quality may be selective. Clear reporting of these steps is critical especially for regulatory purposes that may require relevance assessments demonstrating fit-for-purpose of the RWD.[16] Additionally, pre-registration is not expected for observational studies but is expected for RCTs. At present, there is uncertainty over whether pre-registration should be mandatory, or at least encouraged, for target trial emulation studies.

Third, it may be useful to understand how published studies have discussed the biases and study limitations that preclude perfect emulation of the idealised hypothetical target trial. Recent findings from the Randomised, Controlled Trials Duplicated Using Prospective Longitudinal Insurance Claims: Applying Techniques of Epidemiology (RCT DUPLICATE) initiative compared results of RCTs with their emulation counterparts and found that they do not necessarily agree because perfect emulation may not be possible.[21] This may be due to, but not limited to, differences in treatment adherence rates between RCTs and real-world practice, partial emulation of eligibility criteria due to database limitations, measurement error in the primary outcome and the challenges associated with emulating a placebo add-on arm. Although not every study applying the target trial emulation framework will be able to make comparisons with results from an actual RCT, it is still crucial for authors to postulate on the factors that may strengthen or qualify their interpretation of their results.

### Research aims and objectives

The aims and objectives of this SR are to investigate the practices of studies applying the target trial emulation framework to evaluate the risk-benefit of interventions. We aim to examine multiple facets of the study process, specifically pre-specification, study design, data source, analysis, results, interpretation, and dissemination.

### Objectives

1. To review how trial emulation studies have designed their target trial specification (eligibility criteria, treatment strategies, treatment assignment, outcomes, time zero and follow-up, causal contrasts, statistical analysis).
2. To review how the causal inference assumptions were justified.
3. To review pre-registration, reporting, and dissemination practices.

4. To review what study limitations and biases in target trial emulation are commonly discussed.

## METHODS
### Database search
The following electronic databases will be searched: Medline (via Ovid), Embase (via Ovid; entries from medRxiv are included), PsycINFO (via Ovid), SCOPUS, Web of Science, Cochrane Library, the ISRCTN registry and ClinicalTrials.gov. Search terms for each database are available in online supplemental table 1. The search will be limited to articles published in English only and no publication date limits will be applied. We anticipate a duration of 1.5 years for this SR (from July 2023 until December 2024).

### Study inclusion and exclusion criteria
Eligible studies must first meet the definition of a target trial emulation study based on Hernán and Robins:[4]
1. Presence of a hypothetical target trial specification which explicitly describes the trial design components of eligibility criteria, treatment strategy, assignment procedure, follow-up, primary outcome definition, the causal contrast of interest, and a statistical analysis plan; and
2. Presence of a target trial emulation specification which explicitly describes how the hypothetical target trial is being emulated. The specifications may be displayed in tabular format[14] though this need not strictly be the case so long as the information can be retrieved from the text; and
3. The study must have used at least one observational dataset to conduct their trial emulation.

Additional inclusion criteria are as follows: (1) meets study definition of target trial emulation study; (2) peer-reviewed reports, protocols (published or archived), study registrations and conference presentations on original research studies that used/plan to use the trial emulation, from all fields of research; and (3) available in English language. Studies which sought to replicate RCT findings with RWD will also be included since the RCT protocol contains all the information needed in a hypothetical target trial specification and the target trial emulation specification may be inferred from the methods section.

Exclusion criteria are: (a) conference abstracts, book chapters, opinions or commentaries, purely qualitative papers, SRs and meta-analyses; (b) conceptual/methodological papers that do not use participant data; and (c) non-human subjects.

The reference sections of the excluded SRs, meta-analyses and methodological papers will be checked to identify eligible papers. Additional articles will also be sourced from studies that cited either Hernán and Robins[4] or a closely associated paper published in the same year by Hernán et al.[15]

### Selection of studies for inclusion in the review
TB and SH will read all titles/and or abstracts retrieved from the database search process and eliminate obviously irrelevant records. The full text of the remaining possibly relevant studies will then be obtained and read, with each record classified as clearly relevant (meets all inclusion criteria), clearly irrelevant or insufficient information to decide. For those cases with insufficient information, corresponding author(s) will be contacted for further information to make a final decision on relevance. In the instance that the full text of a possibly relevant study cannot be retrieved, the corresponding author(s) will be contacted and if no reply is received, the record will be excluded from the final review. Disagreements on eligibility of studies will be arbitrated by a third co-author. References will be managed on Rayyan.[22]

### Data extraction and management
TB and SH will extract data from all included records according to a bespoke data capture form (DCF) developed in Excel and designed around the key research objectives. The DCF will be designed to help review co-authors elicit the required information from the study text with specific questions and prompts. Pre-defined responses will be available for all straightforward yes/no-type questions and certain multiple-choice questions. The multiple-choice questions with pre-defined responses will be those in which the investigators have prior knowledge on a list of possible responses; lists that are judged to be non-exhaustive will be supplemented by an 'Other' response. For questions without this prior knowledge, we will ask the co-author extracting the information to copy the appropriate text verbatim from the paper. The text data will then be subjected to a coding process (see Synthesis of qualitative data section). The extracted data will take the form of close-ended choices (eg, yes/no and other pre-defined responses), numerical data (eg, sample size) and qualitative data.

V.1.0 of the DCF will be developed by adapting information from existing reporting frameworks that are judged to be applicable to target trial emulation. These are :
1. CONSORT,[18]
2. STROBE (combined checklist)[19]
3. REporting of studies Conducted using Observational Routinely-collected health Data (RECORD),[23]
4. The estimand framework,[20 24]
5. Risk Of Bias In Non-randomised Studies—of Interventions (ROBINS-I) assessment tool[25] and
6. The STRengthening Analytical Thinking for Observational Studies (STRATOS) publications on initial data analysis,[26] missing data[27] and causal inference.[28]

V.1.0 will then be piloted on 10 of the included papers and adjustments made where appropriate via discussion with additional authors (JW and DT). V.2.0 will then be used for all papers, including the 10 used during the piloting. Key study features to be extracted are outlined in online supplemental table 2; the table functions as a

starting point for data collection, which is an evolving process as will be described by version controls of the DCF.

## Synthesis of qualitative data

Qualitative data may constitute a mixture of brief commentary or discussion points as well as richer qualitative data describing processes and experiences. These data will be analysed thematically drawing on guidelines for synthesising qualitative research in SRs[29] consisting of a three-stage process: (1) TB and SH will independently code each line of extracted text based on content and meaning; (2) initial codes will be grouped together according to similarities to form descriptive themes; and (3) analytic themes will be developed based on interpretation of these data in the context of the aims of this review. Each stage of this process will be undertaken by TB and SH independently followed by discussion to seek agreement of code and theme generation. Where agreement cannot be reached, an additional co-author will arbitrate.

## Summarising and reporting of results

The current protocol adheres to the Preferred Reporting Items for Systematic reviews and Meta-Analyses–Protocol (PRISMA-P) reporting guidelines.[30] Our SR will adhere to the PRISMA reporting guidelines.[31 32]

We will describe the frequency (%) of studies displaying each categorical descriptor of a study feature. For example, when describing types of bias discussed, we may report the frequency and percentage of studies with the following descriptors: 'None stated', 'Unmeasured confounding', 'Selection bias' and so on.

## Risk of bias assessment

No risk of bias assessment will be carried out. It is not within the scope of our objectives to render a judgement on whether the hypothetical target trial was successfully emulated or judge if the study results are credible evidence to inform the effectiveness of a treatment. The evidence being synthesised will primarily be the characteristics of the studies, methods and reporting patterns, as opposed to their results.

## Patients and public involvement

No patient involved.

## Conclusion

This SR of target trial emulation studies will give future researchers a detailed overview of how target trial emulation studies have been conducted and reported. Understanding the current strengths and weaknesses of how current studies are being reported will help to improve the design of trial emulation studies and inform bespoke reporting guidelines.

**Acknowledgements** We thank Marian Rixham from the Newcastle University Walton library for their expertise in optimising the systematic search strategy.

**Contributors** Design of the protocol—TB and SH; draft of the manuscript—TB and SH; review and final approval of the manuscript—TB, SH, JW, DT, AB and MB.

**Funding** TB is funded by the National Institute for Health and Care Research (NIHR) Applied Research Collaboration North East and North Cumbria (Grant/Award Number: NIHR200173). SH (Pre-Doctoral Fellowship, NIHR302746) and JW (Research Professorship, NIHR301614) are funded by the National Institute for Health Research (NIHR) for this research project.

**Disclaimer** The views expressed in this publication are those of the author(s) and not necessarily those of the NIHR, National Health Service (NHS), or the UK Department of Health and Social Care.

**Competing interests** None declared.

**Patient and public involvement** Patients and/or the public were not involved in the design, or conduct, or reporting, or dissemination plans of this research.

**Patient consent for publication** Not applicable.

**Provenance and peer review** Not commissioned; externally peer reviewed.

**ORCID iDs**
Theophile Bigirumurame http://orcid.org/0000-0003-2387-4918
Shaun Kuan Wei Hiu http://orcid.org/0000-0003-1699-4348
M Dawn Teare http://orcid.org/0000-0003-3994-0051
James M S Wason http://orcid.org/0000-0002-4691-126X
Andrew Bryant http://orcid.org/0000-0003-4351-8865
Matthew Breckons http://orcid.org/0000-0003-3057-6767

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
