## [Reviewer comments · BMJ Open]

ARTICLE DETAILS

TITLE (PROVISIONAL)	Current practices in studies applying the target trial emulation framework: A protocol for a systematic review
AUTHORS	Bigirumurame, Theophile; Hiu, Shaun; Teare, Dawn; Wason, James; Bryant, Andrew; Breckons, Matthew

VERSION 1 – REVIEW

REVIEWER	Lavery, Jessica Memorial Sloan Kettering Cancer Center, Epidemiology and Biostatistics
REVIEW RETURNED	25-Jan-2023

GENERAL COMMENTS	The authors present a protocol for a systematic review of current practices in target trial emulation. While randomized clinical trials represent the gold standard for drawing causal inference, appropriate analysis of observational data under the target trial emulation framework may be also provide valuable inference when implemented appropriately. The authors propose to systematically review the literature and document current practices in target trial emulation. This systematic review will provide valuable information regarding the current landscape of target trial emulation protocols and publications that can be used to inform future work in this important area of research. Table 1 • Table 1 oscillates between specifying whether something was stated (e.g., Was the research objective clearly stated?) and documenting the response to something that should be clearly stated (e.g., "What was the population of interest?" rather than "Was the population of interest clearly stated?") Both are likely important. Recommend authors review Table 1 to see if both types of questions need to be incorporated for some of the data items.• The goal of "Were simulations performed?" is not clear as simulations are not always required as part of a sample size justification. Suggest, "Were several scenarios considered for sample size calculations (e.g., varying assumed standard deviation to assess impact on detectable effect size)"• What is the relevance of the cost of the data?• Under dissemination, whether or not the dissemination was peer-reviewed should be listed. Methods: • If your institution offers library services, I highly recommend working with a librarian to conduct the systematic review, if you were not already planning to do so.• Study Inclusion Criteria: By restricting to trial emulations that specify trial design components (eligibility, treatment strategy,
--

	assignment procedure, etc.), you will inherently be missing any trial emulation papers that do not specify all of these components of the hypothetical target trial specification, which seems to be part of the point of the systematic review.  • Data Coding: What you mean by codes is not clear. Are you referring to numeric or categorical codes? Consider expanding this section to clarify. • Consider clarifying how the SR will potentially be used to inform a set of risk of bias tools for target trial emulation. It may be helpful to perform a risk of bias assessment of the studies included in the SR to begin to develop such tools. Table 2:  • Author and year of publication are under sample characteristics, but should be in a separate section of publication/protocol characteristics, which should also include the source of the paper (i.e., which journal, website, etc.) • More demographic features should be added under sample characteristics (e.g., race, ethnicity) • What is the distinction between hospital/GP (GP not listed under acronyms table) and claims? Does hospital/GP refer to data arising from the electronic health record? • Propensity score model is listed under the Results row of the Entire Study column; however, a propensity score model may not always be the analytic method applied, consider broadening. • Whether the material was peer-reviewed should be listed under dissemination. • Currently, the difference between Table 1 and Table 2 is not clear. Table 2 is more granular and complete and contains the information listed on Table 1.
--	--

REVIEWER	Yeo, Anthony Western Sydney University
REVIEW RETURNED	26-Jan-2023

GENERAL COMMENTS	Dear Authors,  * I note that the word "propensity" was used. However many studies do not necessarily report this statistical measure. Findings from various studies can be offered as a linear regression graph/equation, as a survival analysis hazard ratios, or a mixed model, etc. Thus, while I do not object to the use of the "propensity", I would suggest a broader statistical term so that readers are not led to think that only studies which have propensity scores or have propensity models are to be examined in this framework. * When applying this paradigm to generate a simulation, the result(s) may prove to be slightly or moderately inaccurate years later. Perhaps the words "predictive validity" can be added somewhere in the manuscript? I leave it to the authors to decide where to place these 2 words. * Although this is not a meta-analysis, it is akin to a meta-analysis. It does sound like a working plan and what is needed is to conduct a study e.g. with 10 published papers to see how it actually works in practice. Further refinements can then be made based on the data. As such, I recommend publication but it can be abbreviated. A manuscript with data can be published as a full paper.
---

REVIEWER	Knapp, Peter York University
-----------------	---------------------------------

REVIEW RETURNED	07-Feb-2023
-------------

GENERAL COMMENTS	This manuscript reports the protocol for a systematic review of an important topic: trial emulation in observational study data. I have points on the SR design and then other points on the way the paper has been written.  1. The distinction between Tables 1 and 2 is less clear than it could be. I don't really understand why Table 1 includes so many research questions: rather, aren't these ways that elements of the included studies can be categorised? 2. Table 2 states that details of the causal contrast will be extracted, but is this feasible? Hernan and Robins (2016) [ref 4] state that ITT analysis is "rarely possible" in observational analysis of emulation data. 3. In the introduction (paragraph 3) the authors list four of the key principles that must be satisfied for causal inferences to be estimated (ie exchangeability, consistency, positivity and non-interference). Given the importance that is given to these items, shouldn't they be included in the features of studies that will be extracted from studies? (And if they *are* currently included, can that be made clearer?) 4. On the four principles stated above, can their meanings be made clear? It's possible that not all readers will know them. Indeed, I looked for reference to the 4 principles in published papers to ensure that I knew them, and was unable to find a definition of the 4th one listed (non-interference). Ref 7 is to a 300+ pages book and I was unable to find an explanation of the meaning of non-interference there. 5. The third principle that is listed (positivity) seems the most problematic for emulating trials' from observational data. It is that there is a "non-zero probability of all levels of treatment for all types of individuals in the population" - it seems to be that this principle will not be met for some medical treatment trials, perhaps due to non-availability of a treatment but more likely due to prescriber preference for a particular form of treatment (which effectively results in its non-availability to some patients in the population). Perhaps this needs expansion. 6. The paper is written clearly but my view is that some of the text is too similar to the cited sources. For example, the two sentences that comprise the 1st paragraph (from "Randomised controlled trials (RCTs) are..." to "...often not eligible for RCTs") are taken almost word for word from the Maringe article (ref 2). Can some of this be re-written to reduce the similarity? If not, perhaps better just to quote from source articles.
--

VERSION 1 – AUTHOR RESPONSE

REVIEWER 1

1. [Table 1] Table 1 oscillates between specifying whether something was stated (e.g., Was the research objective clearly stated?) and documenting the response to something that should be clearly stated (e.g., "What was the population of interest?" rather than "Was the population of interest clearly stated?") Both are likely important. Recommend authors review Table 1 to see if both types of questions need to be incorporated for some of the data items.

Response: We thank the reviewer for their suggestion. We have removed Tables 1 from the main text but both questions will be considered when we will be extracting data. For instance, if the answer to "Was the population of interest clearly stated?" is yes, we will proceed to answer the question "What was the population of interest?".

2. [Table 1] The goal of "Were simulations performed?" is not clear as simulations are not always required as part of a sample size justification. Suggest, "Were several scenarios considered for sample size calculations (e.g., varying assumed standard deviation to assess impact on detectable effect size)".

Response: We thank the reviewer for this comment. After internal discussion, we removed any questions related to sample size, since they may confuse the reader. Sample size calculations are not necessary for trial emulation (Hernán, 2022).

Hernán MA. Causal analyses of existing databases: No power calculations required. *Journal of Clinical Epidemiology*. 2022;144:203-5.

3. [Table 1] What is the relevance of the cost of the data?

Response: Thank you for highlighting this. The original intention was to delineate between databases that were publicly available versus those that require licensing or fees. We thought that this would be useful to research groups who may not have the requisite budgeting in place to cost for datasets. But upon reflection, collection of this type of data is prone to error as there may be paywalls that are not immediately obvious until one has had personal correspondence with the organisation that owns the data. We have removed this from our objectives.

4. [Table 1] Under dissemination, whether or not the dissemination was peer-reviewed should be listed.

Response: Thank you. We do plan to delineate between whether the publication was published in a peer-reviewed journal vs. Uploaded onto a non-peer-reviewed repository. This is now clarified in Supplemental Table 2 (Dissemination row).

5. [Methods] If your institution offers library services, I highly recommend working with a librarian to conduct the systematic review, if you were not already planning to do so.

Response: Thank you for this advice. We have approached the library services for assistance, and they have helped us to optimise our search terms for each database and the search strategy. We have included them in our acknowledgments. We have included Supplemental Table 1 containing our full search terms for each database.

Additionally, we have revised the search strategy to source for eligible articles that may have followed the target trial emulation framework but did not explicitly call themselves a trial emulation study in either the title or abstract. This will be done by searching through articles that have cited the original Hernán and Robins (reference 4) paper or a closely associated paper by Hernán et al (reference 15) published in the same year (see last paragraph of Study Inclusion and exclusion criteria section).

6. [Methods] Study Inclusion Criteria: By restricting to trial emulations that specify trial design components (eligibility, treatment strategy, assignment procedure, etc.), you will inherently be missing any trial emulation papers that do not specify all of these components of the hypothetical target trial specification, which seems to be part of the point of the systematic review.

Response: This is a very good point. We had anticipated this problem from the outset. It is true that there will be papers that only provide partial information on the components of a hypothetical trial. The eligibility criteria we have put forward is indeed somewhat conservative. But this is for good reason. At the present moment, there is no official definition for what constitutes a trial emulation. So in a very loose sense, any observational study can be viewed as a (however flawed) target trial emulation study.

The area of trial emulation is evolving, and we do expect that the framework set by Hernán and Robins to be something the field will increasingly adopt going forward given its utility in overcoming common biases in comparative effectiveness research with observational data. Reviewing the papers that follow the framework will align our results and conclusions with where the field is going. So perhaps what our review can offer is to obtain the frequency of all studies claiming to be a trial emulation study. From there, we would be able to report the proportion who do follow the framework by Hernán and Robins for whom we will systematically review.

Additionally, we have revised the search strategy to source for eligible articles that may have followed the target trial emulation framework but did not explicitly call themselves a trial emulation study in either the title or abstract. This will be done by searching through articles that have cited the original Hernán and Robins (reference 4) paper or a closely associated paper by Hernán et al (reference 15) published in the same year (see last paragraph of Study Inclusion and exclusion criteria section).

7. [Methods] Data Coding: What you mean by codes is not clear. Are you referring to numeric or categorical codes? Consider expanding this section to clarify.

Response: Thank you. We have clarified the coding to mean assigning a categorical descriptor that best summarises the text (qualitative) data. We are now collaborating with a mixed methodologist (MB) who has provided expert guidance and clarified this section of the methods (see Synthesis of qualitative data section).

8. [Methods] Consider clarifying how the SR will potentially be used to inform a set of risk-of-bias tools for target trial emulation. It may be helpful to perform a risk of bias assessment of the studies included in the SR to begin to develop such tools.

Response: This is an excellent point. However, the risk of bias assessments requires subject matter knowledge. Given that we will potentially be including studies from a wide range of medical specialties e.g., oncology, neurology, psychiatry, cardiovascular, etc, it would not be possible for us to provide valid risk of bias assessments. We decided to remove all points related to “inform[ing] a set of risk bias tools.”

9. [Table 2] Author and year of publication are under sample characteristics, but should be in a separate section of publication/protocol characteristics, which should also include the source of the paper (i.e., which journal, website, etc.)

Response: Thank you. We have moved Table 2 to Supplemental Table 2. We do plan to delineate between whether the publication was published in a peer-reviewed journal vs. uploaded onto a non-peer-reviewed repository. This has been clarified in the Dissemination row of Supplemental Table 2.

10. [Table 2] More demographic features should be added under sample characteristics (e.g., race, ethnicity)

Response: Thank you to the reviewer for raising this. The review comment has prompted further reflection. We have decided to remove the demographic features from the data collection. It is for two reasons – one conceptual and one practical.

The conceptual reason is that our review focuses specifically on the target trial emulation methodology. Even if sample characteristics were extracted, we are not certain how these would aid in the interpretation of the methodology.

The practical reason is that because we are reviewing literature from a wide array of specialties, it is likely that the type of sample characteristics reported will differ substantially. One could presume age and sex to be commonly reported but race, education, smoking, etc. will be dependent on the clinical area.

11. [Table 2] What is the distinction between hospital/GP (GP not listed under acronyms table) and claims? Does hospital/GP refer to data arising from the electronic health record?

Response: Apologies for the confusing text. Yes, that is exactly what we had hoped to capture. We have the idea of organising papers between 'routinely collected data' - a very broad term including administrative claims, registry, and epidemiological surveillance programs (Nicholls et al., 2017), and data collected for research purposes (e.g., longitudinal research cohorts). In more recent years, there will also be the case that research cohorts may be linked with electronic health records (EHR) data, so depending on what we observe in the data, there is likely further classification such as "research cohort-linked with primary care data" for example. We have used the term "Type of observational data" in the Data source row of Supplemental Table 2 to be as broad as possible.

Nicholls SG, Langan SM, Benchimol EI. Routinely collected data: the importance of high-quality diagnostic coding to research. *Cmaj*. 2017 Aug 21;189(33):E1054-5.

12. [Table 2] Propensity score model is listed under the Results row of the Entire Study column; however, a propensity score model may not always be the analytic method applied, consider broadening.

Response: We thank the reviewer for this comment. We have used a different term to capture a wide range of methods that can be used ("Method to mimic randomisation") in the Assignment procedure row of Supplemental Table 2.

13. [Table 2] Whether the material was peer-reviewed should be listed under dissemination.

Response: Thank you. We have incorporated this suggestion in Supplemental Table 2 (Dissemination row).

14. [Table 2] Currently, the difference between Table 1 and Table 2 is not clear. Table 2 is more granular and complete and contains the information listed on Table 1.

Response: We thank the reviewer for their suggestion. We have decided to remove Table 1 and have instead summarised it as a list of research objectives. We removed Table 1 as the list of research questions we could answer from the data was far greater than what we originally put in the table. Thus, to avoid the issue of having to potentially pre-specify a large number of research questions, we have summarised them as objectives to capture the primary motivations for this SR.

Table 2 has been moved to Supplemental Table 2.

REVIEWER 2

1. I note that the word "propensity" was used. However many studies do not necessarily report this statistical measure. Findings from various studies can be offered as a linear regression graph/equation, as a survival analysis hazard ratios, or a mixed model, etc. Thus, while I do not object to the use of the "propensity", I would suggest a broader statistical term so that readers are not led to

think that only studies that have propensity scores or have propensity models are to be examined in this framework.

Response: We thank the reviewer for this suggestion. We have removed table 1 and moved Table 2 to Supplemental Table 2. We have amended the item in the Assignment procedure row of Supplemental Table 2 to reflect a broader term ("Method to mimic randomisation") as propensity score methods are part of a broader class of methods to achieve balance between treatment groups.

2. When applying this paradigm to generate a simulation, the result(s) may prove to be slightly or moderately inaccurate years later. Perhaps the words "predictive validity" can be added somewhere in the manuscript? I leave it to the authors to decide where to place these 2 words.

Response: This is an excellent point. We can never be certain how accurate the emulation of a trial will be.

We have thought about the point the reviewer is bringing up during the planning stages. From an epistemic point of view, it may be challenging for study authors to expand on a prediction or even gauge the degree to which their emulation will be an accurate one without results from actual trials. Some of these target trials may never even come to existence because trial emulation allows one to "simulate" a trial that is not feasible to conduct for example in rare diseases with slow progression.

We thought we could probe at this by collecting information on how the authors have contextualised their findings by discussing the design limitations and potential biases limiting perfect emulation of the hypothetical trial. These are now explicitly listed as objectives and will be extracted (see Discussion row of Supplemental Table 2).

3. Although this is not a meta-analysis, it is akin to a meta-analysis. It does sound like a working plan and what is needed is to conduct a study e.g. with 10 published papers to see how it actually works in practice. Further refinements can then be made based on the data. As such, I recommend publication but it can be abbreviated. A manuscript with data can be published as a full paper.

Response: Thank you to the reviewer for this very pragmatic suggestion. However, we feel it is essential to publish the protocol first.

Publication of a pilot paper implies that data extraction was performed whilst a protocol was not in place. This may put us at odds with the PROSPERO guidelines which do not accept protocols if any sort of data extraction has already begun.

We have proposed to work in a stepwise approach to assessing the feasibility of the data collection with a plan for version controlling of our data collection form. But this will be performed after the publication of the protocol paper. We are also working with a mixed methodologist who will be in a strong position to work with the extracted text (qualitative) data.

REVIEWER 3

1. The distinction between Tables 1 and 2 is less clear than it could be. I don't really understand why Table 1 includes so many research questions: rather, aren't these ways that elements of the included studies can be categorised?

Response: We thank the reviewer for their suggestion. We have decided to remove Table 1 and have summarised it as a list of research objectives. We removed Table 1 as the list of research questions we could answer from the data was far greater than what we originally put in the table. Thus, to avoid the issue of having to potentially pre-specify a large number of research questions, we have summarised them as objectives to capture the primary motivations for this SR.

Table 2 has been moved to Supplemental Table 2.

2. Table 2 states that details of the causal contrast will be extracted, but is this feasible? Hernan and Robins (2016) [ref 4] state that ITT analysis is "rarely possible" in observational analysis of emulation data.

Response: Thank you for raising this. The reviewer is correct in saying that ITT is near impossible in observational studies because in most instances, there would never be an indicator for "treatment assignment." Rather, the closest indicator for this 'observational analogue' of the ITT effect would be "treatment initiation/prescription." This is consistent with Hernan and Robins' statement: "An intention-to-treat analysis, however, is rarely possible in observational analyses of existing data. In our example, the closest observational analogue of the intention-to-treat analysis is a comparison of initiators of the different treatment strategies, assuming adequate adjustment for baseline confounders." We have rephrased the items in Supplemental Table 2 to be "observational analogues" of the ITT and PP effects.

3. In the introduction (paragraph 3) the authors list four of the key principles that must be satisfied for causal inferences to be estimated (ie exchangeability, consistency, positivity and non-interference). Given the importance that is given to these items, shouldn't they be included in the features of studies that will be extracted from studies? (And if they *are* currently included, can that be made clearer?)

Response: Thank you. That is a very good point. We will collect these in our data collection process. We have clarified this in Supplemental Table 2 (see Analysis row and Discussion row).

4. On the four principles stated above, can their meanings be made clear? It's possible that not all readers will know them. Indeed, I looked for reference to the 4 principles in published papers to ensure that I knew them, and was unable to find a definition of the 4th one listed (non-interference). Ref 7 is to a 300+ pages book and I was unable to find an explanation of the meaning of non-interference there.

Response: We thank the reviewer for this comment. We have amended the introduction to include more details regarding the 4 principles as well as the Appendix. Moreover, we have amended Ref 7 to clarify where these principles are explained (chapter 3). Non-interference was explained under "Interference" in Chapter 1, which was indeed confusing. To counter this, we have also included additional references [9- 13] which expand upon all 4 assumptions in detail for an unfamiliar audience.

5. The third principle that is listed (positivity) seems the most problematic for emulating trials' from observational data. It is that there is a "non-zero probability of all levels of treatment for all types of individuals in the population" - it seems to be that this principle will not be met for some medical treatment trials, perhaps due to the non-availability of treatment but more likely due to prescriber preference for a particular form of treatment (which effectively results in its non-availability to some patients in the population). Perhaps this needs expansion.

Response: The reviewer is correct when they say positivity may not always be met in target trial emulation, and that is what we hope to capture in this review.

Using an example of a binary treatment, structural positivity may be violated when for example an investigator includes patients who are contraindicated for one the treatments under investigation. And so within the strata of patients with this contraindication, we will only see one type of treatment. The obvious recommended approach is to exclude patients with contraindications.

Random positivity violations are situations where whilst it is possible to see the variation in treatment within a strata, because of sampling variability there is no variation in that strata in the observed data. This could arise e.g. when sample size is insufficient to accommodate the number of covariates being adjusted for. Some recommendations would be to use overlap weights derived from propensity scores to down-weight patients who fall within the strata without the treatment variation.

We have included an expansion of the four assumptions in the appendix to avoid a verbose introduction.

6. The paper is written clearly but my view is that some of the text is too similar to the cited sources. For example, the two sentences that comprise the 1st paragraph (from "Randomised controlled trials (RCTs) are..." to "...often not eligible for RCTs") are taken almost word for word from the Maringe article (ref 2). Can some of this be re-written to reduce the similarity? If not, perhaps better just to quote from source articles.

Response: Thank you for raising this with us. We have revised the opening paragraph of the introduction.

VERSION 2 – REVIEW

REVIEWER	Lavery, Jessica Memorial Sloan Kettering Cancer Center, Epidemiology and Biostatistics
REVIEW RETURNED	02-May-2023

GENERAL COMMENTS	Thank you to the authors for the revised protocol. It reads well and several points have been clarified and strengthened from the prior version. I commend the authors on this effort, and I am looking forward to the eventual results. I only had two outstanding clarifications:  • Methods/Selection of studies for inclusion in the review: How will papers that reference the Hernan and Robins paper, but do not meet the three-point definition of a target trial emulation study be classified in terms of clearly relevant/irrelevant? Same question for an observational study that is replicating an RCT. The authors may consider changing the selection of studies for inclusion in the review from studies that meet all inclusion criteria to studies that meet at least one inclusion criteria. • Methods/Data extraction and management: What is the difference between version 1.0 and 2.0 of the DCF? Will any papers be reviewed based on v1.0? It is understandable that the DCF may evolve, but ultimately, all papers should be reviewed using the final version of the DCF. Minor comment  • Methods/Database search: The sentence "The start date will be from publication of this protocol." is duplicated, and the following sentence is duplicated with a variation in the duration specified ("We anticipate a duration of one-year for this systematic review" and "We anticipate a duration of 1.5-years for this systematic review.")
---

REVIEWER	Yeo, Anthony Western Sydney University
REVIEW RETURNED	25-Apr-2023

GENERAL COMMENTS	This paper is better written than the one reviewed a few months ago. It seems that any proposed RCT simulation is likely to be therapeutic area specific. For example, a simulation of RCTs of drugs used for the treatment of gout is going to be different from a
---

	simulation for vaccines used against malaria. Thus, I agree with the word "bespoke". On page 6, line 31, it is stated that at least one observational source would be used. An observational source could include 2 studies? I recommend that the authors take into account the trial results of at least 2 studies i.e. 2 datasets and not 1 study to test drive this model in the first instance. After the initial study has been conducted, further refinements to this schema can be taken. The proposed simulation of RCTs has an analogy in meta-analysis. I think that the authors could make a statement on how their results could/would differ from a meta-analysis theoretically and of course practically when they do implement and publish a study. In particular, why any investigator could not simply use a meta-analysis software program to do their analyses instead of using their proposed algorithm. It may be that the author's proposed methodology offers more precise results (because they match also on additional or more qualitative factors as well as quantitative factors)? I am throwing out suggestions as this is a work in progress.
--	--

VERSION 2 – AUTHOR RESPONSE

Reviewer: 1

1. Methods/Selection of studies for inclusion in the review: How will papers that reference the Hernan and Robins paper, but do not meet the three-point definition of a target trial emulation study be classified in terms of clearly relevant/irrelevant? Same question for an observational study that is replicating an RCT. The authors may consider changing the selection of studies for inclusion in the review from studies that meet all inclusion criteria to studies that meet at least one inclusion criteria.

Response: For papers that reference Hernan and Robins, we will then examine the full-text to see if they meet the eligibility criteria. So in effect, we make our final decision based on a full-text scan.

Regarding the second point, we foresee that there may be feasibility and theoretical issues if we do not use all three criteria collectively. Regarding feasibility, from prior knowledge, criterion #2 (presence of the trial emulation specification) will be available by default simply from the methods section. This creates a problem because if we do not set all three points as our eligibility criteria then every observational study then becomes a trial emulation.

From a theoretical point of view, our aim is to review the trial emulation method proposed by Hernán and Robins, which explicitly state that one should detail the hypothetical trial one wishes to emulate. If we departed from this definition, then it becomes questionable if we are even reviewing the trial emulation method.

However, we empathise with the reviewer. It may not be the perfect solution, but it is, we believe, a logically consistent solution that abides by the literature. We do hope that in time, papers declaring themselves as a trial emulation will follow the three criteria.

2. Methods/Data extraction and management: What is the difference between version 1.0 and 2.0 of the DCF? Will any papers be reviewed based on v1.0? It is understandable that the DCF may evolve, but ultimately, all papers should be reviewed using the final version of the DCF.

Response: Thank you for raising this. This was an error on our part. It should read as Version 1.0 will be piloted on 10 papers to inform a final version 2.0. The last point is very sensible. We have now

added that data from papers used in designing the early DCF versions will be re-extracted using the final DCF version.

3.Methods/Database search: The sentence "The start date will be from publication of this protocol." is duplicated, and the following sentence is duplicated with a variation in the duration specified ("We anticipate a duration of one-year for this systematic review" and "We anticipate a duration of 1.5-years for this systematic review.")

Response: Thank you for alerting us to this. We have amended the sentence.

Reviewer: 2

1. It seems that any proposed RCT simulation is likely to be therapeutic area specific. For example, a simulation of RCTs of drugs used for the treatment of gout is going to be different from a simulation for vaccines used against malaria. Thus, I agree with the word "bespoke".

Response: Thank you for raising this. Yes, this is true. The purpose of an emulation is to obtain a causal effect estimate using observational data in lieu of conducting an actual RCT. This may be for investigations into comparative effectiveness and safety, drug repurposing, etc. So the results of a emulation of a trial of, say, Metformin (a treatment for diabetes) versus sulfonylureas (a class of antidiabetic medication) would only generalise to an RCT of Metformin versus sulfonylureas. The results of an emulation of a trial comparing gout treatments would not generalise an emulation of malaria vaccines. But this is not to say that the methodology used in one trial emulation cannot be used across multiple clinical areas. For example, the use of 'nested trials' (a method of trial emulation) has been used in emulating trials of Statins vs. Placebo, atorvastatin vs. simvastatin (Danaei et al., 2018), and levosimendan vs. placebo (Massol et al., 2023).

Danaei G, Rodríguez LA, Cantero OF, Logan RW, Hernán MA. Electronic medical records can be used to emulate target trials of sustained treatment strategies. *Journal of clinical epidemiology*. 2018 Apr 1;96:12-22.

Massol J, Simon-Tillaux N, Tohme J, Hariri G, Dureau P, Duceau B, Belin L, Hajage D, De Rycke Y, Charfeddine A, Lebreton G. Levosimendan in patients undergoing extracorporeal membrane oxygenation after cardiac surgery: an emulated target trial using observational data. *Critical Care*. 2023 Feb 7;27(1):51.

2.On page 6, line 31, it is stated that at least one observational source would be used. An observational source could include 2 studies? I recommend that the authors take into account the trial results of at least 2 studies i.e. 2 datasets and not 1 study to test drive this model in the first instance. After the initial study has been conducted, further refinements to this schema can be taken.

Response: We apologise for creating confusion. The purpose of this criterion was simply to exclude studies where an "emulation" was conducted purely from clinical trial data (e.g., a single arm trial's data was used as the intervention arm, and a control arm is taken from a different trial with the same inclusion and exclusion criteria) – which would defeat the purpose of this methodology's power to capitalise on existing observational data. We have now amended this point to make it clearer.

3.The proposed simulation of RCTs has an analogy in meta-analysis. I think that the authors could make a statement on how their results could/would differ from a meta-analysis theoretically and of course practically when they do implement and publish a study. In particular, why any investigator could not simply use a meta-analysis software program to do their analyses instead of using their proposed algorithm. It may be that the author's proposed methodology offers more precise results (because they match also on additional or more qualitative factors as well as quantitative factors)? I am throwing out suggestions as this is a work in progress.

Response: Thank you for raising this. We appreciate the suggestions that the reviewer is providing. If we have understood the reviewer's comment, they are raising the issue of the relative gains of a trial emulation, particularly the use of causal inference methodology, versus a meta-analysis of already-conducted studies.

We think this is an important clarification, but perhaps one that, strategically, would be better placed in the final paper. Once a reader has had a chance to reflect on the strengths and weaknesses of trial emulation from our results, discussion on its relative merits would be appreciated.

The causal effect estimates of a series of trial emulations of, say, drug A vs. B can themselves be meta-analysed. So perhaps it may be more advantageous to view trial emulation and meta-analyses as cooperative methods instead of competing methods whom a researcher has to decide which to use. This is good scientific practice because even with the promise of trial emulation, we can never be certain that any emulation is a perfect representation of an RCT's results, and so we reduce our uncertainty through a meta-analysis. We quote Miguel Hernán who puts this into perspective most succinctly: "The solution cannot be refusing to carry out each of those observational analyses with imprecise estimates, but rather to encourage the conduct of many such observational analyses. After several studies become available, we can meta-analyze them and provide a more precise pooled effect estimate... When a causal question is important, it is preferable to have multiple studies with imprecise estimates than having no study at all."

Hernán MA. Causal analyses of existing databases: no power calculations required. Journal of clinical epidemiology. 2022 Apr 1;144:203-5.

Reviewer: 3

1.The authors have responded to the peer review points and the paper is clearer now.

Response: Thank you for reviewing our manuscript.

VERSION 3 – REVIEW

REVIEWER	Lavery, Jessica Memorial Sloan Kettering Cancer Center, Epidemiology and Biostatistics
REVIEW RETURNED	24-May-2023

GENERAL COMMENTS	Thank you to the authors for their work on this revision. The protocol is clear and reads well, and I have no further comments. I'm looking forward to the results of this work.
--

REVIEWER	Yeo, Anthony Western Sydney University
REVIEW RETURNED	27-May-2023

GENERAL COMMENTS	Dear Authors, On page 6, line 31, the authors state that they will use 1 database. Just add at least 1 sentence to describe the database, the content that it should contain so that a simulation can be conducted. This can clear up any possible confusion to a reader. I look forward to the publication of your first paper with this method. This is a method in evolution.
--

	Kind regards, A reviewer
--	-----------------------------

VERSION 3 – AUTHOR RESPONSE

Reviewer: 1

1. Thank you to the authors for their work on this revision. The protocol is clear and reads well, and I have no further comments. I'm looking forward to the results of this work

Response: Thank you for reviewing our manuscript.

Reviewer: 2

1. On page 6, line 31, the authors state that they will use 1 database. Just add at least 1 sentence to describe the database, the content that it should contain so that a simulation can be conducted. This can clear up any possible confusion to a reader.

Response: We thank the reviewer for their comment and we apologise for the confusion. It was not our intention to give readers the impression that we will be using any sort of observational database. Rather, it is the papers that we are going to review that have used the observational database(s) to conduct their trial emulation study.

However, we understand that a description of these database(s) are important and so we will be describing the observational database(s) that these papers have used as part of our results to help inform the conduct of future trial emulations.

We would like to clarify that we are neither using any observational data to conduct a trial emulation nor are we conducting any form of a simulation.

To make this clearer, we have amended the following:

- o [Page 6, line 19] “Eligible studies must first meet the definition of a target trial emulation study based on Hernán & Robins [4]. A study will be included if the following three criteria are met:”
- o [Page 6, line 31] “The study must have used at least one observational dataset to conduct their trial emulation.”
- o [Page 6, line 32] “Additional inclusion criteria are as follows...”